# HOMOTOPY-BASED TRAINING OF NEURALODES FOR ACCURATE DYNAMICS DISCOVERY

## ABSTRACT

Conceptually, Neural Ordinary Differential Equations (NeuralODEs) pose an attractive way to extract dynamical laws from time series data, as they are natural extensions of the traditional differential equation-based modeling paradigm of the physical sciences. In practice, NeuralODEs display long training times and suboptimal results, especially for longer duration data where they may fail to fit the data altogether. While methods have been proposed to stabilize NeuralODE training, many of these involve placing a strong constraint on the functional form the trained NeuralODE can take that the actual underlying governing equation does not guarantee satisfaction. In this work, we present a novel NeuralODE training algorithm that leverages tools from the chaos and mathematical optimization communities – synchronization and homotopy optimization – for a breakthrough in tackling the NeuralODE training obstacle. We demonstrate architectural changes are unnecessary for effective NeuralODE training. Compared to the conventional training methods, our algorithm achieves drastically lower loss values without any changes to the model architectures. Experiments on both simulated and real systems with complex temporal behaviors demonstrate NeuralODEs trained with our algorithm are able to accurately capture true long term behaviors and correctly extrapolate into the future.

## 1    INTRODUCTION

Predicting the evolution of a time varying system and discovering mathematical models that govern it is paramount to both deeper scientific understanding and potential engineering applications. The centuries-old paradigm to tackle this problem was to either ingeniously deduce empirical rules from experimental data, or mathematically derive physical laws from first principles. However, the complexities of the systems of interest have grown so much that these traditional approaches are now often insufficient. This has led to a growing interest in using machine learning methods to infer dynamical laws from data.

One school of thought, such as the seminal work of Schmidt & Lipson (2009) or Brunton et al. (2016), focuses on deducing the exact symbolic form of the governing equations from data using techniques such as genetic algorithm or sparse regression. While these methods are powerful in that they output mathematical equations that are directly human-interpretable, they require prior information on the possible terms that may enter the underlying equation. This hinders the application of symbolic approaches to scenarios where there is insufficient prior information on the possible candidate terms, or complex, nonlinear systems whose governing equations involve non-elementary functions.

On the other hand, neural network-based methods, such as Raissi et al. (2018), leverage the universal approximation capabilities of neural networks to model the underlying dynamics of the system without explicitly involving mathematical formulae. Of the various architectual designs in literature, Neural Ordinary Differential Equations(NeuralODEs) Chen et al. (2018) stand out in particular because these seamlessly incorporate neural networks inside ordinary differential equations (ODES), thus bridging the expressibility and flexibility of neural networks with the de facto mathematical language of the physical sciences. Subsequent works have expanded on this idea, including blending NeuralODEs with partial information on the form of the governing equation to produce "grey-box"

dynamics model (Rackauckas et al., 2021), and endowing NeuralODEs with mathematical structures that the system must satisfy (Greydanus et al., 2019; Finzi et al., 2020).

However, despite the conceptual elegance of NeuralODEs, training these models tend to result in long training times and sub-optimal results, a problem that is further exacerbated as the length of the training data grows (Ghosh et al., 2020; Finlay et al., 2020). Different methods have been proposed to tackle the problem, but majority of these approaches to date involve placing either strong (Choromanski et al., 2020; Hasani et al., 2021), or semi-strong constraints(Finlay et al., 2020; Kidger et al., 2021) to the functional form the NeuralODE can take - something the underlying governing equation does not guarantee satisfying.

**Contributions.** We introduce a novel training algorithm that does not require architectural constraints to accurately train NeuralODEs on long time series data. As our algorithm is inspired by ideas from the chaos and mathematical optimization literature, we provide a background survey on the ideas involved before providing both a general framework and a specific implementation for our algorithm. Experiments on various systems of difficulties demonstrate that our method always outperforms conventional gradient-descent based training, with resulting trained NeuralODEs having both higher interpolation and extrapolation capabilities than their counterparts. Especially, for the relatively simple Lotka-Volterra system, we report a $\times 10^2$ improvement for interpolation error and a staggering $\times 10^7$ improvement in extrapolation error, showcasing the power of our new approach.

## 2 BACKGROUND

### 2.1 NEURAL ORDINARY DIFFERENTIAL EQUATIONS

A NeuralODE (Chen et al., 2018) is a model of the form,

$$\frac{d\boldsymbol{u}}{dt} = \boldsymbol{U}(t, \boldsymbol{u}; \boldsymbol{\theta}), \quad \boldsymbol{u}(t = t_0) = \boldsymbol{u}_0 \tag{1}$$

where $\boldsymbol{u}_0 \in \mathbb{R}^n$ is the initial condition or input given to the model, and $\boldsymbol{U}(...; \boldsymbol{\theta}) : \mathbb{R} \times \mathbb{R}^n \to \mathbb{R}^n$ is neural network with parameters $\boldsymbol{\theta} \in \mathbb{R}^m$ that governs the dynamics of the model state $\boldsymbol{u} \in \mathbb{R}^n$ over time $t \in \mathbb{R}$. The value of the model state at a given time can then be evaluated by numerically integrating equation 1 starting from the initial conditions.

In this paper, we concern ourselves with the problem of training NeuralODEs on time series data. Specifically, given an monotonically increasing sequence of time points $\{t^{(i)}\}_{i=0}^N$ and the corresponding vector-valued measurements $\{\hat{\boldsymbol{u}}^{(i)} \in \mathbb{R}^n\}_{i=0}^N$, we wish to train a NeuralODE on the data to learn the underlying governing equation and forecast future data.

Conventionally, NeuralODE training starts with using an ordinary differential equation (ODE) solver to numerically integrate equation 1 to obtain the model state $\boldsymbol{u}$ at given time points :

$$\{\boldsymbol{u}^{(i)}(\boldsymbol{\theta})\}_{i=0}^N = \text{ODESolve}\left(\frac{d\boldsymbol{u}}{dt} = \boldsymbol{U}(t, \boldsymbol{u}; \boldsymbol{\theta}), \{t^{(i)}\}_{i=0}^N, \boldsymbol{u}_0\right) \tag{2}$$

with $\boldsymbol{u}^{(i)}(\boldsymbol{\theta})$ being a shorthand for $\boldsymbol{u}(t^{(i)}; \boldsymbol{\theta})$. Afterwards, the loss function $\mathcal{L}(\boldsymbol{\theta}) : \mathbb{R}^m \to \mathbb{R}$ is computed according to

$$\mathcal{L}(\boldsymbol{\theta}) = \frac{1}{N+1} \sum_i l\left(\boldsymbol{u}^{(i)}(\boldsymbol{\theta}) - \hat{\boldsymbol{u}}^{(i)}\right) \tag{3}$$

where $l(\boldsymbol{u}, \hat{\boldsymbol{u}})$ is the pairwise loss function. In this paper, we adopt the widely used mean-squared error function $l(\boldsymbol{u}, \hat{\boldsymbol{u}}) = ||\boldsymbol{u} - \hat{\boldsymbol{u}}||^2/n$, but other metrics such as the L1 loss can be used (Finzi et al., 2020; Kim et al., 2021).

Training is performed by minimizing equation 3 via gradient descent. A non-trivial aspect of this process is that computing $\nabla_{\boldsymbol{\theta}}\mathcal{L}$ requires differentiating the ODESolve operation. This can be done by either directly backpropagating through the internals of the ODE solver algorithm - which returns accurate gradients but is memory intensive - or by the "adjoint method", which computes an auxiliary set of ODEs to obtain gradients at a low memory cost, but can yield inaccurate gradients. In this paper, we embrace recent advances in the field and use the "symplectic-adjoint method", which brings the best of both worlds by having both low memory footprint and improved accuracy guarantees (Matsubara et al., 2021).

## 2.2 SYNCHRONIZATION OF DYNAMICAL SYSTEMS

Consider the following two dynamical systems,

$$\frac{d\boldsymbol{u}}{dt} = \boldsymbol{U}(t, \boldsymbol{u}; \boldsymbol{\theta}), \quad \frac{d\hat{\boldsymbol{u}}}{dt} = \hat{\boldsymbol{U}}(t, \hat{\boldsymbol{u}}; \hat{\boldsymbol{\theta}}) \tag{4}$$

where $\boldsymbol{u}, \hat{\boldsymbol{u}} \in \mathbb{R}^n$ are $n$-dimensional state vectors, and $\boldsymbol{U}, \hat{\boldsymbol{U}} : \mathbb{R} \times \mathbb{R}^n \to \mathbb{R}^n$ are the dynamics of the two systems with coefficients $\boldsymbol{\theta}, \hat{\boldsymbol{\theta}}$. The functional form of the first system $\boldsymbol{U}(...; \boldsymbol{\theta})$ is known and we wish to fit this to data generated from the second system, $\{\hat{\boldsymbol{u}}^{(i)}\}_{i=0}^N$, at time points $\{t^{(i)}\}_{i=0}^N$. We assume $\boldsymbol{U} = \hat{\boldsymbol{U}}$, that is, the functional form of the data generating dynamics is the same as the equation to be fitted and the initial conditions of the two systems are identical.

As the coefficients $\boldsymbol{\theta}, \hat{\boldsymbol{\theta}}$ of the two systems are different, the resulting trajectories $\boldsymbol{u}(t), \hat{\boldsymbol{u}}(t)$ will, in general, be independent of each other. For chaotic systems, the result is more drastic as the solutions to these systems are extremely sensitive to parameter variations (Figure 1). For the periodic system (left panel), we find that a $\pm 50\%$ parameter perturbation only changes the period and amplitude of the signal. In contrast, perturbation of the same relative magnitude on a chaotic system (middle panel) results in completely different behaviors for the resulting solution.

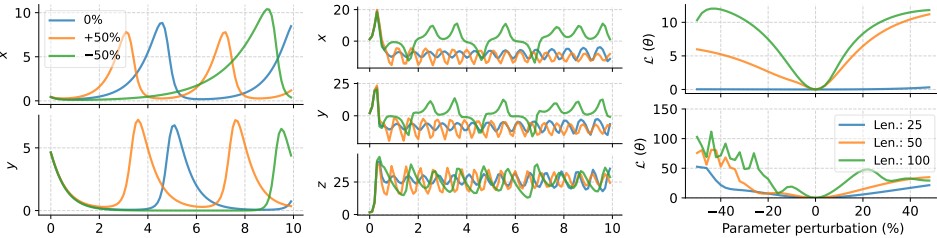

Figure 1: Dynamics of single parameter perturbed systems. The systems and the parameters used are described in Experiments. (**Left**) Solutions for the periodic Lotka-Volterra equations with perturbed $\alpha$ parameter. (**Middle**) Solutions for the chaotic Lorenz system with perturbed $\beta$ parameter. (**Right**) MSE loss function landscape for the Lotka-Volterra equations (Up) and Lorenz system (Down) for different lengths of time series data for the loss calculation.

Determining the coefficients $\boldsymbol{\theta}$ of the known equation $\boldsymbol{U}$ to find the unknown coefficients $\hat{\boldsymbol{\theta}}$ underlying the data $\{\hat{\boldsymbol{u}}^{(i)}\}_{i=0}^N$, proceeds analogously to the NeuralODE training described in section (2.1): the equation to be fitted is numerically solved for a given parameter guess, the loss function is calculated with respect to data, then minimized using gradient descent to yield the final answer. However, the independent evolution of two uncoupled dynamics can easily lead to non-convex loss functions with sharp local minima, especially for longer, more irregular time series (Figure 1, right panel). This, in turn, leads to unstable loss function minimization and finally resulting in inaccurate coefficients (Voss et al., 2004). Furthermore, this problem of irregular loss function during training also occurs in recurrent neural network (RNN) and NeuralODE training as well (Doya, 1993; Ribeiro et al., 2020).

Enter a slightly altered version of equation 4, which has an additional term in the $\boldsymbol{u}$ dynamics that couples the two systems:

$$\frac{d\tilde{\boldsymbol{u}}}{dt} = \tilde{\boldsymbol{U}}(t, \tilde{\boldsymbol{u}}; \boldsymbol{\theta}) = \boldsymbol{U}(t, \tilde{\boldsymbol{u}}; \boldsymbol{\theta}) - \boldsymbol{K}(\tilde{\boldsymbol{u}} - \hat{\boldsymbol{u}}), \quad \frac{d\hat{\boldsymbol{u}}}{dt} = \hat{\boldsymbol{U}}(t, \hat{\boldsymbol{u}}; \hat{\boldsymbol{\theta}}) \tag{5}$$

where the $\boldsymbol{K} = diag(k_1, ..., k_n)$ is the diagonal coupling matrix $\boldsymbol{K}$. It was found that if the elements of $\boldsymbol{K}$ are positive and are sufficiently large, the dynamics of $\boldsymbol{u}$ and $\boldsymbol{v}$ synchronize with increasing time regardless of the parameter mismatch: meaning $\|\tilde{\boldsymbol{u}}(t) - \hat{\boldsymbol{u}}(t)\|_2 \to 0$ as $t \to \infty$ (Abarbanel et al., 2009; Pecora & Carroll, 2015). Figure 2 illustrates this phenomenon for both periodic (left panel) and chaotic (middle panel) systems, using $\boldsymbol{K} = k\boldsymbol{I}$ where $k \in \mathbb{R}^+$ is the scalar coupling strength and $\boldsymbol{I}$ is the $n \times n$ identity matrix.

The new coupling term acts similarly to the proportional control term used in PID control, and actively drives the modified dynamics $\tilde{\boldsymbol{u}}(t)$ towards the reference trajectory $\hat{\boldsymbol{u}}(t)$. Increasing the

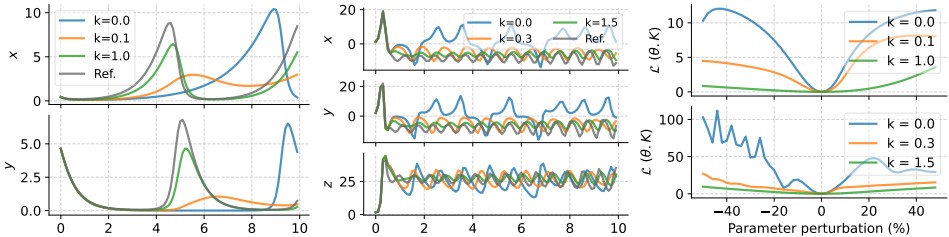

Figure 2: Dynamics of coupled systems with varying coupling strengths. The systems and the parameters used are identical to those of figure 1. **(Left)** Results for the periodic Lotka-Volterra equations. **(Middle)** Results for the chaotic Lorenz system. **(Right)** Loss function landscape of the coupled systems with different coupling strengths.

coupling strength correspond to turning up the proportional gain, and causes the two trajectories to converge even faster (Figure 2, left and middle panels).

Repeating the parameter estimation procedure, the loss function $\mathcal{L}$ between $\tilde{u}$ and $\hat{u}$ now becomes a function of both $\theta$ and $K$ ($= kI$) (Figure 2, right panel). In the context of loss function calculation, increasing the coupling strength $k$ effectively acts as shortening the length of the data because increased coupling causes the two trajectories to start synchronizing earlier in time, after which the error between the two will diminish and contribute negligibly to the total loss. Therefore, synchronizing the model and the data generating equations leads to a more favorable loss landscape, which gradient descent based optimizers will not struggle on.

However, directly minimizing this new loss function $\mathcal{L}(\theta, K)$ associated with the coupled system (equation 5) does not result in our desired goal of discovering the unknown data generating coefficients $\hat{\theta}$. This is because our goal requires us to fit the original uncoupled equation (4) on the data, and not the modified equation (5) that has an additional term. Therefore, a method that reaps the benefits of a better-behaved loss function using synchronization, but also results in a minimizer of the original loss function $\mathcal{L}(\theta)$ is needed. That method is homotopy optimization, which we describe in the following section.

### 2.3 HOMOTOPY OPTIMIZATION AND PARAMETER IDENTIFICATION

Consider the problem of finding the parameter vector $\theta^* \in \mathbb{R}^m$ that minimizes the function $\mathcal{F}(\theta)$ : $\mathbb{R}^m \to \mathbb{R}$. While minimization algorithms, such as the commonly used gradient descent methods, can directly be applied to the target function, these tend to struggle if $\mathcal{F}$ is non-convex and has a complicated landscape riddled with local minima. To address such difficulties, the homotopy optimization method (Dunlavy & O'Leary, 2005) introduces an alternative objective function

$$\mathcal{H}(\theta, \lambda) = \begin{cases} \mathcal{G}(\theta), & if \quad \lambda = 1 \\ \mathcal{F}(\theta), & if \quad \lambda = 0 \end{cases} \tag{6}$$

where $\mathcal{G}(\theta) : \mathbb{R}^m \to \mathbb{R}$ is an auxillary function whose minimum is easily found, and $\mathcal{H}(\theta, \lambda) : \mathbb{R}^m \times \mathbb{R} \to \mathbb{R}$ is a continuous function that smoothly interpolates between $\mathcal{G}$ and $\mathcal{F}$ as the homotopy parameter $\lambda$ is varied from 1 to 0.

The motivation behind this scheme is similar to simulated annealing: one starts out with a relaxed version of the more complicated problem of interest, and finds a series of approximate solutions while slowly morphing the relaxed problem back into its original non-trivial form. This allows the optimization process to not get stuck in spurious sharp minima and accurately converge to the minimum of interest.

To proceed with the method, one first selects the number of discrete steps for optimization, $s$, as well as a series of positive decrement values for the homotopy parameter $\{\Delta\lambda^{(k)}\}_0^{s-1}$ that sum to 1. Afterwards, optimization starts with an initial $\lambda$ value of $\lambda^{(0)} = 1$, which gives $\mathcal{H}^{(0)}(\theta) = \mathcal{G}(\theta)$. At each step, the objective function at the current iteration is minimized with respect to $\theta$, using the

output from the previous step $\boldsymbol{\theta}^{*(k-1)}$ is as the initial guess, and :

$$\mathcal{H}^{(k)}(\boldsymbol{\theta}) = \mathcal{H}(\boldsymbol{\theta}, \lambda = \lambda^{(k)}) \ \rightarrow \ \boldsymbol{\theta}^{*(k)} = \underset{\boldsymbol{\theta}}{\arg \min} \ \mathcal{H}^{(k)}(\boldsymbol{\theta}) \tag{7}$$

Afterwards, $\lambda$ decremented to its next value, $\lambda^{(k+1)} = \lambda^{(k)} - \Delta\lambda^{(k)}$, and this iteration continues until the final step $s$ where $\lambda^{(s)} = 0$, $\mathcal{H}^{(s)}(\boldsymbol{\theta}) = \mathcal{F}(\boldsymbol{\theta})$, and the final minimizer $\boldsymbol{\theta}^{*(s)} = \boldsymbol{\theta}^*$ is the sought-after solution to the original minimization problem $\mathcal{F}(\boldsymbol{\theta})$.

**Application to synchronization.** To combine homotopy optimization with synchronization, we slightly modify the coupling term of equation 5 by multiplying it with the homotopy parameter $\lambda$:

$$- \boldsymbol{K}(\tilde{\boldsymbol{u}} - \hat{\boldsymbol{u}}) \quad \rightarrow \quad -\lambda \boldsymbol{K}(\tilde{\boldsymbol{u}} - \hat{\boldsymbol{u}}). \tag{8}$$

With this modification, applying homotopy optimation to the problem is straightforward. When $\lambda = 1$ and the coupling matrix $\boldsymbol{K}$ has sufficiently large elements, synchronization occurs and the resulting loss function $\mathcal{L}(\boldsymbol{\theta}, 1 \cdot \boldsymbol{K})$ is well-behaved, serving the role of the auxillary function $\mathcal{G}$ in equation 6. When $\lambda = 0$, the coupled equation 5 reduces to the original equation 4, and the corresponding loss function $\mathcal{L}(\boldsymbol{\theta}) = \mathcal{L}(\boldsymbol{\theta}, 0 \cdot \boldsymbol{K})$ is the complicated loss function $\mathcal{F}$ we need to ultimately minimize. Therefore, starting with $\lambda = 1$ and successively decreasing its value to 0 in discrete steps, all the while optimizing for the coupled loss function $\mathcal{L}(\boldsymbol{\theta}, \lambda\boldsymbol{K})$ allows one to leverage the well-behaved loss function landscape from synchronization while being able to properly uncover the system parameters (Vyasarayani et al., 2012; Schäfer et al., 2019).

## 3 HOMOTOPY OPTIMIZATION FOR NEURALODE TRAINING

While the approaches in the previous sections were developed for the problem of determining the unknown coefficients of a differential equation from data, we propose to apply them to the problem of training a NeuralODE on time series data. This conceptual leap is based on two observations:

1. The previous formalism on synchronization and homotopy optimization can directly be translated over to NeuralODE training by reinterpreting $\boldsymbol{U}(..., \boldsymbol{\theta})$ from 'a known equation whose coefficients $\boldsymbol{\theta}$ we wish to find' to 'a NeuralODE with parameters $\boldsymbol{\theta}$ we wish learn the data generating equation $\hat{\boldsymbol{U}}(..., \boldsymbol{\theta})$ with.

2. While subsection 2.2 considered the case where the model equation differs only in coefficients with the data generating equation ($\boldsymbol{U} = \hat{\boldsymbol{U}}$), it has been discovered that two systems with different functional forms can also synchronize - a phenomena called "generalized synchronization" (Pecora & Carroll, 2015). This motivates our coupling a NeuralODE to a given time series data to facilitate training.

### 3.1 IMPLEMENTATION DETAILS

The previous sections describe a general methodology of using homotopy optimization for NeuralODE training. However, there are couple details that need addressing in order to create a specific implementation of our algorithm. We briefly discuss these points below.

**Construction of the coupling term** In the previous section, the coupling term $-\boldsymbol{K}(\tilde{\boldsymbol{u}} - \hat{\boldsymbol{u}})$ in the right hand side of equation 5 was implicitly thought to be defined for all time $t$. However, NeuralODEs are trained on measurements $\{\hat{\boldsymbol{u}}^{(i)}\}_{i=0}^{N}$ sampled at discrete time points $\{t^{(i)}\}_{i=0}^{N}$, which makes the control term undefined on any other time points. Leaving the control term partially defined is not an option, as most ODE solver algorithms require evaluating the right hand side of equation 5 at intermediate time points as well. Therefore, one must extend the definition of the control term so that it is defined even at times where the measurements $\hat{\boldsymbol{u}}^{(i)}$ are not supplied.

One solution to this problem is to apply the coupling term as a sequence of impulses, only applying it where it is defined, and keeping it zero at other times. While this is an interesting avenue of interest - especially so since this can be shown to be mathematically equivalent to teacher forcing(Quinn et al., 2009), a method used to mitigate instabilities during RNN training(Toomarian & Barhen, 1992) - we do not adopt this method in this paper. Instead, we construct smoothing cubic splines

from the data points to supply the measurement values at unseen time points that are needed to evaluate the coupling term, with smoothing chosen over interpolation to make our approach robust to measurement noise in the data.

**Scheduling the homotopy parameter**  While the homotopy parameter $\lambda$ can be decremented in various different ways, the most common approach is to use constant decrements: $\Delta\lambda^{(k)} = 1/s$, $k \in [0, s-1]$ where $s$ is the number of homotopy steps. In our study, we modify this slightly so that the decrements are decayed by a constant ratio for each passing step (Figure 3, left panel) - that is,

$$\Delta\lambda^{(k+1)} = \kappa_\lambda \Delta\lambda^{(k)}; \quad \sum_{k=0}^{s-1} \Delta\lambda^{(k)} = 1; \quad 0 < \kappa_\lambda \leq 1. \tag{9}$$

### 3.2 Algorithm overview

Our final implementation of the homotopy training algorithm has five hyperparameters. Here, we briefly describe the effects and the tips for tuning each.

- **Number of homotopy steps** ($s$) : This determines how many relaxed problem the optimization process will pass through to get to the final solution. Similar to scheduling the temperature in simulated annealing, fewer steps results in the model becoming stuck in a sharp local minima, and too many steps makes the optimization process unnecessarily long. We find using values in the range of 6-8 or slightly larger values for more complex systems yields satisfactory results.

- **Epochs per homotopy steps** ($n_{epoch}$) : This determines how long the model will train on a given homotopy parameter value $\lambda$. Too small, and the model lacks the time to properly converge on the loss function; too large, and the model overfits on the simpler loss landscape of $\lambda \neq 0$, resulting in a reduced final performance when $\lambda = 0$. We find for simpler monotonic or periodic systems, values of 100-150 work well; for more irregular systems, 200-250 are suitable.

- **Coupling strength** ($k$) : This determines how trivial the auxillary function for the homotopy optimization will be. Too small, and even the auxillary function will have a jagged landscape; too large, and the initial auxillary function will become flat (Figure 2, left panel, $k = 1.0, 1.5$) resulting in very slow parameter updates. We find good choices of $k$ tend to be comparable to the scale of the measurement values.

- **$\lambda$ decrement ratio** ($\kappa_\lambda$) : This determines how the homotopy parameter $\lambda$ is decremented. Values close to 1 cause $\lambda$ to decrease in nearly equal decrements, whereas smaller values cause a large decrease of $\lambda$ in the earlier parts of the training, followed by subtler decrements later on. We empirically find that $\kappa_\lambda$ values of 0.5-0.6 tends to work well.

- **Learning rate** ($\eta$) : This is as same as in conventional NeuralODE training. We found values in the range of 0.01-0.1 to be adequate for our experiments.

## 4 Experiments

We evaluate the performance of our algorithm by training NeuralODEs on three different systems of varying difficulties selected from literature: the Lotka-Volterra system, the Lorenz equations and the double pendulum. In both homotopy and baseline experiments, the MSE loss was monitored during training, and the checkpoint corresponding to this MSE minimum was used for further analysis. Each experiment was repeated three times with different random seeds, and we report the corresponding means and standard errors as solid lines and shaded bands wherever applicable. Additional details regarding the experiments can be found in the **??**.

### 4.1 Lotka-Volterra system

We start our experiments with the Lotka-Volterra system, which is a simplified model of predator-prey dynamics given by the following equations,

$$\frac{dx}{dt} = \alpha x - \beta xy, \quad \frac{dy}{dt} = -\gamma y + \delta xy \quad (10)$$

where $x(t), y(t)$ denote the prey and predator populations as a function of time, and $\alpha, \beta, \gamma, \delta$ are parameters regarding the interaction between the populations (Murray, 2002). We follow the experimental design of Rackauckas et al. (2021), where the goal of the experiment is to fit the following hybrid model on the simulated data,

$$\frac{dx}{dt} = \alpha x + U_1(x, y; \boldsymbol{\theta}_1), \quad \frac{dy}{dt} = -\gamma y + U_2(x, y; \boldsymbol{\theta}_2) \quad (11)$$

where $U_1(x, y; \boldsymbol{\theta}_1), U_2(x, y; \boldsymbol{\theta}_2)$ are neural networks employed to discover the corresponding "unknown" terms $-\beta xy, \delta xy$ from the given data.

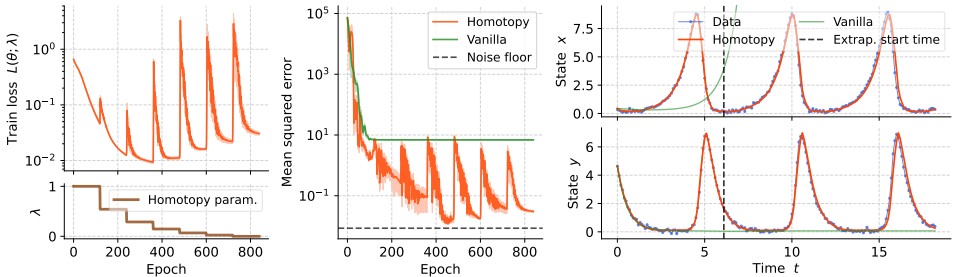

Figure 3: Lotka-Volterra dataset results. **(Left)** Train loss $\mathcal{L}(\boldsymbol{\theta}, \lambda)$ and homotopy parameter $\lambda$ during training. **(Middle)** Mean squared error as a function of training epochs for both homotopy-based and vanilla NeuralODE. Models corresponding to the minimum of this curved are used to plot the prediction results on the right. **(Right)** Interpolation and extrapolation results for both the homotopy and vanilla trained models. The dashed gray line indicates the end of training data and the start of extrapolation.

Figure 3 displays results for both vanilla and our homotopy-based NeuralODE training. We find that not only does our training method reach the final MSE value of the vanilla NeuralODE in about half the number of epochs, the best MSE error achieved is near to the noise floor of the dataset - which is the lowest value the MSE can take without model overfitting. Inspecting Figure 3 (right), we find NeuralODEs trained with the homotopy method is able to perfectly fit the training data in the interpolation regime as suggested by extremely low best MSE value of the middle panel. The high performance of the trained model continues far into the extrapolation regime (after the dashed gray line), where it near perfectly predicts the data across multiple future periods. As the final trained model was selected only using the training data and had not seen any data in the extrapolation regime, the near perfect future predictions indicate that our training method accurately extracted the underlying dynamics of the data. These results are in stark contrast to those obtained by standard NeuralODE training, whose results overfit the $y$ variable and neglect the $x$ variable almost entirely and has no predictive capabilities.

**Training on different length trajectories.** Simply training NeuralODEs with a gradient-based optimizer gives drastically deteriorating results as the length of training trajectories increases. To illustrate this point, simulation trajectories of different time spans were generated: $t \in [0, 3.1], [0, 6.1], [0, 9.1], \Delta t = 0.1$, and the same hybrid model (equation 11) was trained using both vanilla and homotopy methods with identical hyperparameter settings as before.

Figure 4 shows the training results for Lotka-Volterra train data of different lengths. In the case of interpolation, our method consistently trains models to loss values near the noise floor, regardless of the length of the training data. On the other hands, the conventional training method only manages to train NeuralODEs for very short data, and completely fails to learn the oscillatory dynamics that start to arise at longer times. The differences are even more drastic for the case of extrapolation, where there is a $\times 10^7$ MSE difference between our approach and the baseline for longer time series. In the case of short data, our result also gives a high, albeit slightly lower loss value than the baseline.

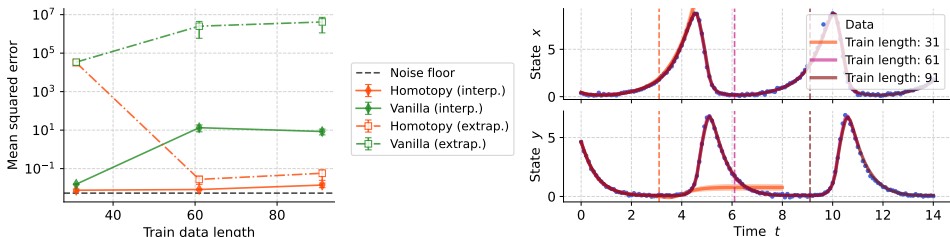

Figure 4: Different train data length results. **(Left)** Interpolation and extrapolation MSE values as a function of training data length. Extrapolation MSE was calculated by computing prediction values for 50 points after the end of training data, then computing the MSE value against the noise-added true dynamics. Error bars indicate one standard error away from the mean over three runs. **(Right)** Interpolation and extrapolation results for models trained with our method. End of training data is marked by the colored dash lines, and extrapolations are performed for 50 points after the end of train data to match the results on the left panel.

Inspecting the right panel, this is revealed to be due to a lack of information in the training data - given access to less than half the oscillation period, it would be impossible for the model to determine whether the true dynamics is oscillatory or not.

## 4.2 LORENZ EQUATIONS

For our next set of experiments, we turn to the Lorenz equations, given by

$$\frac{dx}{dt} = \sigma(y - x), \quad \frac{dy}{dt} = x(\rho - z) - y, \quad \frac{dz}{dt} = xy - \beta z. \tag{12}$$

The chaotic nature of these equations makes even the seemingly simple task of determining the unknown coefficients of the equations from the data extremely nontrivial. Hence, the Lorenz equations serve as the perfect stress test for training a NeuralODE on the data.

The task of the experiment was to fit the "black-box" model of equation 1 on the simulated and noise added trajectory of 31 points, and the resulting extrapolation performance of the trained models were inspected using another 31 points following the end of training data.

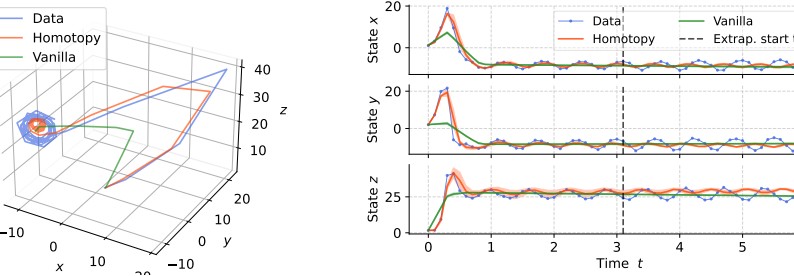

Figure 5: Lorenz system: **(Left)** State space trajectories for the train data and learned dynamics using the homotopy and vanilla training methods. **(Right)** Interpolation and extrapolation results on the Lorenz equations dataset for both homotopy and vanilla trained results.

Figure 5 again highlights the effectiveness of our homotopy training against the baseline approach. Without homotopy, the NeuralODE fails to learn any useful feature of the system dynamics, collapsing into a flat line after the first few points in the interpolation interval. In contrast, results from the homotopy training is able to accurately fit the training data in the interpolation interval, having learnt both the initial peak and the subsequent oscillations. Due to the complexity of the chaotic system, the model predictions does start to go out-of-sync with the ground truth in the extrapolation regime, but it is evident the dynamics learned with homotopy is much richer and accurate than its baseline counterpart.

### 4.3 DOUBLE PENDULUM

For our final set of experiments, we used real-world measurement data of the dynamics of a double pendulum taken by Schmidt & Lipson (2009). The goal of the experiment was to fit another "black-box" model on the training data, which consists of 100 time points, and monitor the model predictions for another 50 points afterwards.

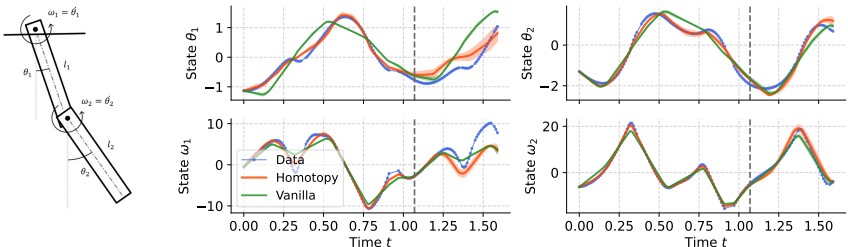

Figure 6: Double pendulum: **(Left)** Diagram of the double pendulum with the state variables $\theta_1$, $\theta_2$, $\omega_1$, $\omega_2$ denoted. $l_1$ and $l_2$ correspond to the lengths of the pendulum bars. **(Right)** Interpolation and extrapolation results on the double pendulum dataset for both homotopy and vanilla trained results.

From Figure 6, we see once again that our homotopy approach produces models that are much more accurate in the interpolation regime than those by the conventional training method. For extrapolation, both homotopy and baseline results fail to accurately predict all four state variables. However, the homotopy result does much closely follow the rises and falls of the ground truth data (especially for $\theta_1$) than its counterpart, demonstrating the effectiveness of our algorithm.

## 5 RELATED WORKS

**Stabilizing NeuralODE training** The necessity for improving NeuralODE training has been noticed by the community, and multiple different methods have been proposed. Works such as Choromanski et al. (2020) or Hasani et al. (2021) use specific mathematical forms for their NeuralODEs that guarantee stability. However, for the purpose of discovering dynamical equations underlying the data, these approaches are inadequate as one cannot ascertain their hidden equations conform to the rigid mathematical structure of these approaches. Another class of methods, such as Finlay et al. (2020) and Kelly et al. (2020) use a softer form of regularization to boost training, by placing constraints on the Jacobian or high derivatives of the differential equation to be learned. However, these methods were also developed in the context of using NeuralODEs for input-output mapping. There, as long as the mapping is the same, the actual form of the dynamics underneath does not matter - which is not the case in our problem setting. Ghosh et al. (2020) introduces a method that does not impose any constraint on the model form, but treats the integration time points as stochastic variables. Other methods such as Kidger et al. (2021), Zhuang et al. (2020), Matsubara et al. (2021), and Kim et al. (2021) study other aspects of NeuralODE training, including better gradient calculation using improved adjoint methods. Such methods are fully compatible with our proposed work, as evidenced by our use of the "symplectic-adjoint-method" of Matsubara et al. (2021) in our experiments.

## 6 DISCUSSION AND OUTLOOK

In this paper, we adapted the concepts of synchronization and homotopy optimization for the first time in the NeuralODE literature, and demonstrated that models trained with our proposed method are unrivaled in both interpolation and extrapolation accuracy across three different systems. As our training algorithm is first of its kind for NeuralODEs, there are diverse avenues of future research, including a more through investigation on the presented hyperparameter selection heuristics, or using regularization terms in loss function to decrease the coupling term as opposed to manual scheduling (Abarbanel et al., 2009; Schäfer et al., 2019). As NeuralODEs are continuous analogs of RNNs, our work holds new prospect for stabilizing RNN training as well.

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

## A  EXPERIMENT DETAILS

In all experiments, the AdamW optimizer (Loshchilov & Hutter, 2017) was used to minimize the respective loss functions for the vanilla and homotopy training. The number of hidden layers was fixed to two for all experiments, with hyperbolic tangent activation for the input and hidden layers, and the identity function for the output layer. Hyperparameters were chosen by running hyperparameter sweeps and selecting the resulting best values.

### A.1  LOTKA-VOLTERRA SYSTEM

Following the experimental design of Rackauckas et al. (2021), we numerically integrate equation 10 in the time interval $t \in [0, 6.1]$, $\Delta t = 0.1$ with the parameter values $\alpha, \beta, \gamma, \delta = 1.3, 0.9, 0.8, 1.8$ and initial conditions $x(0), y(0) = 0.44249296, 4.6280594$. Continuing with the recipe, Gaussian random noise with zero mean and standard deviations with magnitude of 5% of the mean of each trajectory was added to both populations. The integration was performed using the dopri5 solver from the torch-symplectic-adjoint package (Matsubara et al., 2021) was used with an absolute tolerance of 1e-9 and a relative tolerance of 1e-7.

The neural networks $U_1(x, y; \boldsymbol{\theta}_1)$ and $U_2(x, y; \boldsymbol{\theta}_2)$ had 2 nodes for the input layer, 5 nodes for its two hidden layers, and 1 node for the output layer. Deviating from Rackauckas et al. (2021), we used hyperbolic tangent activation functions instead of the radial basis function activation that the previous authors used. This change was made to better adhere to the standard practices of the NeuralODE community.

## A.2  LORENZ ATTRACTOR

To generate the data, we followed experimental settings of Vyasarayani et al. (2012), we used the parameter values $\sigma, \rho, \beta = 10, 28, 8/3$ and the initial condition $x_0, y_0, z_0 = 1.2, 2.1, 1.7$ which, upon integration, gives rise to the well-known "butterfly attractor". Adhering to the paper, a Gaussian noise of mean 0 and standard deviation 0.25 was added to the simulated trajectories to model experimental noise. The ODE solver and the tolerance values were kept identical to the previous Lotka-Volterra experiment.

The NeuralODE used for the experiment had 50 nodes for its two hidden layers, and 3 nodes for its input and output layers corresponding to the three degrees of freedom of the state vector $\boldsymbol{u} = [x, y, z]^T$ of the system.

## A.3  DOUBLE PENDULUM

The experimental data from Schmidt & Lipson (2009) consists of two trajectories of the double pendulum, captured using multiple cameras. The noise in the data is subdued due to the LOESS smoothing performed by the original authors. For our experiments, we used the first 100 points of the first trajectory for training and the next 50 for evaluating the extrapolation capabilities of the trained model.

Similar to the previous experiment, we used a "black-box" NeuralODE with 50 nodes for the hidden layers, and the input and output layer nodes changed to 4 to reflect the four degrees of freedom of the system (Figure 6, left panel).

