# OpenReview forum: "Homotopy-based training of NeuralODEs for accurate dynamics discovery"
_ICLR.cc/2023/Conference — Submitted to ICLR 2023_

### Official Review · Reviewer_4PH8 · 2022-10-20

**Confidence:** 3
**Correctness:** 3
**Technical Novelty And Significance:** 1
**Empirical Novelty And Significance:** 2
**Recommendation:** 3

**Clarity, Quality, Novelty And Reproducibility:**

**Quality:**
The presented experiments are not sufficient enough to be conclusive. One major limitation is that the proposed method is only compare against one baseline, which is vanilla gradient descent. I do appreciate that the hyper parameters are roughly indicated. However, I believe that the hyper parameters for all methods must be optimization, which is not described in the manuscript.

At parts the presentation is not optimal. Sometimes, the manuscript is not very concise, for example, Subsection 2.2 could be significantly shortened without sacrificing clarity. Occasionally, some unfinished sentences, missing space and linguistic mistakes are present. At times, the tone of writing is rather informal, which I personally do not appreciate.

The discussion of related works is present, but not sufficient. Most importantly, it omits the literature on parameter identification for dynamical systems, which is a closely related task to the problem of regression of time series data with neural ODEs.

**Repreducability:** The manuscript does provide most implementation details, but does not make the code available.

**Clarity:**
Overall, the ideas presented in the manuscript are easy to follow and mostly clearly presented.

**Originality:**
The idea of using homotopy optimization is very intriguing but not original. It has been applied in different works for parameter identification in dynamical systems. Note that neural ODEs can be viewed as an instant of a general parameter estimation problem in a dynamical system. Whereas some works from the literature on parameter identification are referenced, I believe they are not credited enough, since they are neither mentioned in the introduction nor the related works section and are therefore hard to miss. Rather, the manuscript claims in the contribution section to introduce a novel optimization algorithm, where I would the manuscript applies the existing method of homotopy optimization with synchronization to train neural ODEs. That being said, I am not aware of papers pursuing this approach in the context of training neural ODEs and believe that strong enough experiments can justify a publication.


**Strength And Weaknesses:**

**Strengths:**
* The idea of using a homotopy to a problem with a more regular landscape to leverage difficulties in the training of neural ODEs is very attractive. Also the demonstration that such a problem can be constructed via synchronization is valuable.

**Weaknesses:**
* Apart from being an attractive idea, homotopy continuation with synchronization has been used for parameter identification in dynamical systems before. Hence, the contribution of the manuscript reduces to the application of an existing method for parameter identification to the special case of neural ODE training.
* Being a purely empirical work, the experiments are not very strong. In particular, the proposed method is only compared against one baseline, which is the vanilla gradient descent training of the neural ODE.

**Summary Of The Paper:**

The manuscript proposes to use homotopy optimization between a stabilized version of a neural ODE and the original neural ODE to learn dynamical systems from time series data. The proposed method is tested on three synthetic problems, where it is shown to outperform a naive gradient descent training of the neural ODE.

**Summary Of The Review:**

The manuscript addresses the important and timely problem of learning dynamical systems from time series data using neural ODEs.
I appreciate the idea of using a homotopy to a sychronized system in order to leverage training of neural ODEs.
Currently, I have the a few main concerns regarding the manuscript in its current form:
* The idea of homotopy optimization with a synchronized system is not new. However, prior works from the parameter identification literature are only mentioned at the very end of Subsection 2.3 and neither in the introduction nor in the related works section. In general I would describe the contribution of the paper rather the application of this approach in the context of neural ODE training as they fall in the abstract setting of parameter identification as far as I can see it (see for example Schäfer et. al. 2019). That being said, I don’t mind applying an idea from another field, but I think the contribution of the manuscript in comparison to prior works have to be sufficiently clear.
* Being an empirical paper, I am currently not convinced that the experiments are comprehensive enough. In particular, the proposed method is only compare against one baseline, which is vanilla gradient descent.
* The overall presentation can be improved.

Overall, I believe that the manuscript is not ready for publication in its current form. Nevertheless, I want to highlight that I find the idea of using homotopy optimization for neural ODE training intriguing and believe that with stronger experiments, this can be an interesting direction.

---

> ### Author Response · Authors · 2022-12-12
> **Thank you to reviewer 5**
>
> Thank you for your comments. While we were not able to update new results in time for this conference, we wanted to acknowledge your comments and let you know that they have helped us a lot in further improving our work.
>
> #1. We agree that homotopy continuation with synchronization alone has been used in the dynamical systems literature. We want to clarify that we are not attempting to present that idea as our own, but rather trying to apply the technique to the scope of NeuralODE training. We have and will make this point clearer in future versions of our work.
>
> #2. Motivated by your comments, while we were not able to obtain the results in time, we have expanded on our experiments, which showed the effectiveness of our algorithms on many different NeuralODE architectural choices. Further investigations on comparing against different methods of training still need to be done, and we will explore this avenue further in our subsequent experiments.
>
> Once again, thank you and we very much appreciate your constructive criticism.

---

> > ### Comment · Reviewer_4PH8 · 2022-12-12
> > **Good luck!**
> >
> > Dear authors,
> >
> > great to hear that some of my feedback might have been helpful. I wish you all the best with this project and would enjoy it being published soon.
> >
> > Best

---

### Official Review · Reviewer_89pi · 2022-10-23

**Confidence:** 3
**Correctness:** 3
**Technical Novelty And Significance:** 4
**Empirical Novelty And Significance:** 2
**Recommendation:** 5

**Clarity, Quality, Novelty And Reproducibility:**

- The manuscript is readable.
- The proposed model is novel.
- The validity of the proposed method is somewhat unclear.

**Strength And Weaknesses:**

S1. This paper is the first example of introducing synchronization and homotopy optimization for learning Neural ODEs.
S2. The effectiveness of the proposed method is verified for multiple physical systems.

W1. The proposed method is novel, however, it is somewhat unclear why the introduction of synchronization and homotopy optimization solves the problem. First of all, can you briefly explain why the learning becomes unstable for long duration data?　Then, can you explain how the proposed method solves the problem? An intuitive introduction would also be helpful.
W2. As the authors said, It is a well-known fact that learning is unstable for long duration data. For practical use, a time-window is introduced and a shortened sequence is used as a mini-batch for training. What are the advantages of the proposed method over such techniques?
W3. Can the proposed method be applied to machine learning tasks (regression, classification, etc.)? If so, it would be good to discuss its effectiveness.

**Summary Of The Paper:**

This paper proposes a new method for training Neural ODEs by employing techniques of synchronization and homotopy optimization. In the task of learning physical systems, the authors have shown that it is possible to learn appropriately for long duration data.

**Summary Of The Review:**

The proposed method is novel, but its validity and effectiveness are somewhat unclear.

---

> ### Author Response · Authors · 2022-12-12
> **Thank you to reviewer 4**
>
> Thank you for your valuable comments. We would like to provide some answers to your questions above.
>
> #1. While we were not able to explain in depth due to the space constraint, the reason why training becomes unstable for longer time series is as follows. When the model is predicting the data trajectory, it only relies on the model dynamics and not the data. Therefore the model prediction evolves independent of the data, and tends to diverge away from the data dynamics as time increases. This leads to loss values increasing rapidly for longer time series data, leading to a more unstable optimization problem. This problem is further obfuscated by bifurcations that can occur during training, which is when the time behavior of the model dynamics changes abruptly even with a incremental change in the model parameters. This leads to exploding gradients, which manifest as stiff, jagged cliffs in the optimization landscape. We provide a toy visualization of this phenomena in Figure 1.
>
> Synchronization and homotopy optimization solves this problem in two ways. First, synchronization constrains the evolution of the model dynamics to be close to the data dynamics, which prevents the two trajectory from diverging away. This leads to smaller, more stable loss values during training. Furthermore, with a suitable coupling coefficient, bifurcation in the model dynamics can be avoided as well (provided that the this coupling coefficient results in a negative conditional maximum Lyapunov exponent value) leading to a very smooth optimization landscape in the more difficult, initial stages of training.
>
> Another intuitive way to think of synchronization is that it effectively reduces the length of the training data, as we mention in the text and in our Figure 2.
>
> #2. Your comment on training on a single long time series versus training on a batch of shorter segments is very much on point. While we have yet to perform detailed experiments, it is very likely that the optimal way of training on long time series data is to split it into moderately long segments and train over a batch of them. The problem of what is the optimal training segment length for a given trajectory is not trivial and requires further investigation, but that was not the scope of the paper in this work. In future works, we aim to apply our method to batch training, as well as determine the optimal segment size for training long time series data.
>
> #3. The structure of our algorithm is most suited to training on time series prediction tasks. To apply them to a input-output mapping problem such as regression or classification are a bit different from our method as they utilize the NeuralODE architecture in different ways.

---

### Official Review · Reviewer_3MeW · 2022-10-23

**Confidence:** 4
**Correctness:** 2
**Technical Novelty And Significance:** 3
**Empirical Novelty And Significance:** 2
**Recommendation:** 3

**Clarity, Quality, Novelty And Reproducibility:**

### Clarity

The paper is well written and clear.

### Quality

The paper develops very intersting ideas but lacks the depth to provide a comprehensive motivation of the approach. In particular, theoretical motivation for the coupling as well as stress tests of the approach to understand when it fails are missing.

### Novelty

The paper builds upon a long line of work in Neural ODEs and optimization. In that regard, I think the title of the paper is slightly off point as it focuses on homotopy training while the core contribution relies on the coupling in my opinion.

### Reproducibility

The descriptions for reproducibility are present in the paper. It's not clear why the Neural ODEs results are so bad in the lokta-voltera case.

**Details Of Ethics Concerns:**

I have no ethical concerns.

**Strength And Weaknesses:**

### Strengths

- The coupling of the dynamics to improve the training is a very nice and promising idea.
- The paper is easy to read and understand

### Weaknesses

In general, I think this paper does not go deep enough either theoretically or practically, therefore missing the impact it could have.

- I believe it would be very interesting to have a more in-depth investigation of the properties of the loss including the coupling. You show graphically that this coupling results in a smoother loss landscape but your work requires more theoretical foundations to motivate the approach in general.  You state that previous works have found that it synchronizes but you should do the extra step of deriving what it implies for the loss.

- From an experimental point of view, it would be desirable to have some real world data as well. It's ok if the theoretical motivation is not strong but then you need a solid experimental setup. Additionally, it seems you are using only single realization of time series. How does your model transfer to the case of multiple realization of a time series (say different clinical trajectories for instance) ?

- In terms of limitations, the fact that coupling is required at every time step is significant. I think it would be interesting to know when the spline interpolation fails, when the sampling rate because lower and lower.  In particular, the sampling rate used in your experiments seems quite high.

- The mean square errors you get from the vanilla Neural ODEs in Figure 4 seem unrealistic and might point towards some regularization issue in the Neural ODE.  It's not clear from your experiments how you used regularization for training. I think it should be fair to have an extensive hyperparameters search to compare against your method. Also, why are the results so bad for the Lokta Voltera but seems to be acceptable for Lorenz and Double Pendulum which are arguably much more difficult systems ?


**Summary Of The Paper:**

This paper proposes a new training paradigm for Neural ODEs by annealing a coupling term between the learnt system and the target system. This coupling allows for a more efficient training of the Neural ODE by simplifying the loss landscape. The authors then showcase their approach on synthetic data (Lorenz, LV and double pendulum).

**Summary Of The Review:**

Interesting and promising idea for improving the training of neural odes. Nevertheless, the paper currently lacks depth in the investigation. In particular, theoretical motivation for the more behaved loss and/or more comprehensive experiments. In its current form, the paper is nor theoretically nor experimentally convincing but I fully believe there is the potential for it to be in the future.

---

> ### Author Response · Authors · 2022-12-12
> **Thank you to reviewer 3**
>
> Thank you for your valuable comments and constructive criticisms. While we were not able to update our work in time, we wanted to acknowledge how your comments have strived us to improve and provide some responses to your questions.
>
> #1. As our paper was very empirical, we do acknowledge that while we did try to provide some theoretical motivations in subsections 2.2 and 2.3, our paper does not build on strong mathematical theories and proofs. This is due to the complexity of the problem, and we will keep trying to tackle that in our future research.
>
> #2. To clarify, the data for the double pendulum, which was one of the three datasets we considered, was real world experimental data and not simulated results. However, we do agree that it would be an interesting avenue to apply our method to more “industrial” data, such as the clinical trajectories you have mentioned. While that is not the scope of this particular research, it is a direction we aim to pursue.
>
> #3. While we were not able to produce the results within time, your comments have led us to investigate the robustness of our algorithm against increased sparsity and noise levels in the data, both of which can degrade the spline interpolant we construct for the coupling. Interestingly we find that it is not the sparsity, but the noise level that our algorithm is more vulnerable to, as increased noise degrades the quality of the spline more drastically than decreased sampling rate. We do believe, however, that this weakness can be addressed, by adapting the smoothing level of the interpolant during construction, to the noise level in the data.
>
> #4. We agree that the loss values reported for the vanilla NeuralODE training for the Lotka-Volterra dataset can seem unrealistic. However, we do want to clarify that this is not due to a regularization issue in our training, but rather due to the extremely small model size (2 hidden layers, 5 nodes per hidden layer) and small number of train epochs (=1000) used. While our homotopy method was still able to train an accurate model under this challenging conditions, the baseline method was unable to, hence the extremely poor prediction results. This was less of an issue for the more difficult Lorenz and double pendulum systems, as the neural networks used for those systems had the same number of hidden layers, but a much larger number of nodes per hidden layer (50 vs 5), resulting in a better fit.
>
> Building on your comment, and to create better baselines, we have performed new experiments with increased number of hidden nodes per layer (5 → 20 for the Lotka-Volterra system, and 50 → 200 for Lorenz and double pendulum systems) and increased number of maximum training epochs (1000 → 4000). This has led to a much better performance for the baseline models, most of them being able to properly fit the data in the interpolation interval. However, we still find that our homotopy training method triumphs, as it requires x2-x6.5 times the number of training epochs to arrive at an accuracy equal to or greater than the baseline results.
>
> Once again, we thank you for your comments. While we weren’t able to update the results in time for this conference, your comments have led to great improvements in our work, to be included in newer versions of our preprint. Thank you.

---

### Official Review · Reviewer_nRZJ · 2022-10-24

**Confidence:** 3
**Correctness:** 4
**Technical Novelty And Significance:** 3
**Empirical Novelty And Significance:** 2
**Recommendation:** 6

**Clarity, Quality, Novelty And Reproducibility:**

Clarity:

The paper is well-written with sufficient details.

Quality:

The proposed method is technically sound. Because the proposed method is simple (principled), I found no concern in the correctness of it.

The experimental results seem to be weak as I mentioned in Weakness section.

Novely:

To me, the proposed homotopy optimization for exploiting the synchronization of dynamical systems is novel. However, my evaluation might not exhaustive because I am not on top of the current literatures.

Reproducibility:

The paper contains experimental details in the main text and supplementary. Code is not made publicly available.


**Strength And Weaknesses:**

Strength:
1. The motivation of the introduced coupling term for synchronizing NODEs and true dynamics is clear and principled. The authors convincely explain the effectiveness of such a coupling term when estimating the optimal parameters of dynamical systems.

2. The coupling-based synchronization cannot be used directly for training NODEs, because it gives an undesired model that depends on the unknown true dynamics. To overcome this issue, the authors introduce a homotopy between the coupled and uncoupled NODE loss functions. To me, the application of the homotopy optimization for training NODEs with the synchronization is novel.

3. The paper is very well-written and easy-to-follow.

Weakness:

1. The proposed method has four additional hyper-parameters that should be tuned carefully. Although the authors provide rough guidelines for tuning such hyper-parameters, it is not entirely clear whether the proposed method is sufficiently robust and general.

2. The experimental results on Lorenz and double pendulum are not sufficiently strong. There are some extrapolation errors. Considering that they are chaotic, it is acceptable (for me). For the double pendulum experiment, it might be interesting to apply the proposed homotopy optimization to Hamiltonian NODEs.

**Summary Of The Paper:**

The authors propose a novel training strategy for neural ordinary differential equations (NODEs) by introducing a coupling term between the true dynamics and NODEs. They formulate the coupling-based training framework by using the homotopy optimization, which optimizes a homotopy between the simple coupled loss landscape and the complicated uncoupled loss landscape. It results in stable and fast training of NODEs with respect to the targeted dynamical system. The authors validate their proposed learning framework for three standard benchmarks (Lotka-Volterra, Lorenz, and double pendulum) that frequently used in NODE literatures.

**Summary Of The Review:**

I think the paper is interesting, as I mentioned above. I would like to vote to (weak) accept for this paper.

---

> ### Author Response · Authors · 2022-12-12
> **Thank you to reviewer 2**
>
> Thank you for your comments and your interest in our work. While we were not able to update our work in time, we did take great note of your comments and wanted to provide some responses to your comments.
>
> #1. We agree that our method does introduce multiple new hyperparameters, which requires some tuning to yield performance. However, the algorithm does not require extensive searching in all five parameters - rather, we find that it is usually sufficient to only search over the learning rate and the coupling strength, and keep other parameters fixed and predetermined values. This was the strategy used in our previous submission, with weights & biases used to perform the two dimensional hyperparameter sweep. Acknowledging your comment, we have further clarified and added additional details as to how the hyperparameters were chosen in our updated preprint, uploaded elsewhere.
>
> #2. The extrapolation errors for the Lorenz and the double pendulum system occurred due to the size of the neural networks used being small (2 hidden layers, 50 hidden nodes each). After the submission, we performed newer experiments with increased number of nodes per layer (50 → 200), which led to better, albeit not perfect, extrapolation qualities. Also, while we are yet to perform experiments for Hamiltonian neural network, we have performed additional experiments using a second-order NeuralODE for the double pendulum. This is done because a recent study has shown that much of the performance gains in HNNs arise from the incorporated second-order structure in the model formulation (N. Gruver et al, ICLR 2021). Results from the additional experiments show that our homotopy training method also excels in training second-order models as well, with the performance boost over baseline training being even more drastic than that of a regular NeuralODE. We are still interested in HNNs as well, and are currently working on experimenting on those models as well.
>
> Once again, we thank you very much for your comments.

---

### Official Review · Reviewer_zjmg · 2022-10-26

**Confidence:** 5
**Correctness:** 3
**Technical Novelty And Significance:** 4
**Empirical Novelty And Significance:** 3
**Recommendation:** 5

**Clarity, Quality, Novelty And Reproducibility:**

The paper is clearly written and offers enough details to reproduce the method. It is novel: it proposes an approach that allows to make a match between the trained ODE and the ground-truth trajectory. These techniques have not been mentioned in the space of Neural ODE literature, to my knowledge.

The authors tested their model on a range of systems, including chaotic, and in the variety of scenarios, for example, with added noise. As mentioned above, I have concerns about the properly trained Neural ODE baseline, but otherwise I am happy with the paper quality.


**Strength And Weaknesses:**

**Strengths**

Both techniques introduced in the paper come from the classic optimisation theory. Intuitively, the idea is similar to simulated annealing — first optimise an easier function that is already close to the ground-truth, and then increase the difficulty of the optimised task.

The paper demonstrates the approach on notoriously challenging systems, such as Lorenz attractor and double pendulum. Since Lorenz attractor is a chaotic system, it requires a very accurate reconstruction of the trajectory and is hard to extrapolate correctly. The coupling term ensures that the trained ODE can accurately fit the target trajectory. Figure 2 also provides an insight on how the optimised function changes because of the coupling term.

**Weaknesses**

1. My main concern is the evaluation of the Neural ODE baselines. It is surprising that Neural ODE has extremely poor fit on Lotka-Volterra model in Figure 3 and Lorenz attractor on Figure 5, but can fit the double pendulum reasonably well (Figure 6).  I trust that the coupling term and the homotopy would help the ODE training (mostly based on results in Figure 6), but it is hard to evaluate the benefits of the proposed approach, if the baseline is not well trained.


    To my understanding, the training is done on one trajectory, so Neural ODE should be able to overfit the training data, or at least have a better fit than just a constant function. For example, in the original paper the model was able to fit the oscillating data: https://arxiv.org/abs/1907.03907. This is the example where Neural ODE is able to fit the trajectory from Lotka-Volterra model: https://gist.github.com/ChrisRackauckas/a531030dc7ea5c96179c0f5f25de9979

    Figures 3 and 5 suggest that the Neural ODE baseline is either under-parameterized, or not trained long enough.


    Can the authors provide a similar experiment to figure 3, where the Neural ODE is able to fit at least the training part of the trajectory, or explain why they think the results are not consistent with the results of the previous papers?


2. The proposed approach uses splines to impute the input data to construct the coupling term. The splines can provide the reasonable interpolation on densely-sampled data with low amounts of noise, but might require a more careful consideration in case of sparse data or data with higher amount of noise (a common scenario in the real-world data). However, this is not a big limitation, because the splines are only used in the coupling term, which is set to zero in the end of the training.

**Minor**
Section 2.1: “Differentiating the ODESolve operation … can be done by the ”adjoint method”, which …. can yield inaccurate gradients.”

Can the authors give examples or citations when the adjoint gradients can be inaccurate, but the ”symplectic-adjoint” integrators are accurate?

**Summary Of The Paper:**

The paper proposes a technique for matching the predicted trajectory to the trained ODE using a Coupling term. It also introduces the idea of Homotopy training, where we have the interpolation between a smooth, easy function and the target function that we want to optimise. Using these two techniques allow to better fit the Neural ODE to the target trajectory, including Lotka-Volterra models, Lorenz attractor and double pendulum. Using these techniques also lead to the better extrapolation of the predicted trajectory.

**Summary Of The Review:**

The paper introduces an interesting technique to enforce the match between the ground-truth trajectory and the. Trained ODE, as well a simulated-annealing-like technique to start the optimisation from an easier function.

However, the results of the baseline (Neural ODE) are not consistent with the results from the previous papers and external sources, suggesting that it was not trained properly or under-parameterised. I am giving the weak-reject, but willing to greatly increase the score if the authors re-train the Neural ODE model on Lotka-Volterra and Lorenz attractor and show the better fit than just predicting a constant function, or explain why their Neural ODE results might be inconsistent with the previous works.

---

> ### Author Response · Authors · 2022-12-12
> **Thank you to the reviewer**
>
> Thank you for taking an interest in our research and providing detailed comments. While we have not been able to provide a revised manuscript addressing your concerns in the allotted time, we wanted to provide rationales and answers.  We have numbered our responses to your comments below, in the order you have mentioned them.
>
> #1. Response to the poor model performance
>
> When we were performing the training for this paper, we were focused on training the baseline and our homotopy approach with the same amount of epochs. The number of training epochs was chosen to be about 1000 -  a number that was sufficient for our homotopy approach, but too small to train a NeuralODE with the vanilla approach. Through additional experiments performed after receiving your comments, we found that at least 4000 epochs were needed to train a NeuralODE on the our experimental systems  with the baseline approach, confirming your guess.
>
> For the Lotka-Volterra dataset, there was another reason for the poor baseline performance, that the size of the NeuralODE used was very small (2 hidden layers, 5 hidden nodes per layer). While this choice was intentionally made to mirror the experimental settings used in C. Rackauckas et al, arXiv:2001.04385 (2021), your comments have made us to realize that this, coupled with the small number of training epochs results in very weak baselines. Similarly, the model for the Lorenz dataset had two hidden layers with 50 nodes each, which also seems to have been underparametrized.
>
> While we have not been able to produce the results in time for this conference, we have performed a new batch of experiments with increased epochs and some architecture changes to the NeuralODEs. This includes increased model size all experimental systems (5 → 20 hidden nodes per layer for the Lotka-Volterra dataset, and 50 → 200 hidden nodes for both Lorenz and double pendulum systems), and change in the activation function from tanh to gelu, an approach proposed in S. Kim et al, Chaos 31:093122 (2021) that we found to be extremely effective in increasing the trainability of NeuralODEs. We find that our homotopy method can train models with similar or better accuracy using x2 - x6.5 times less training epochs depending on the dataset.
>
> Furthermore, we have also trained a model akin to the one provided in your github link (single hidden layer, 32 nodes per hidden layer, no prior information about the equation given), and found that our method was still effective in that case.
>
> #2. Effect of noise and sparsity on the spline interpolation and algorithm performance
>
> Based on your comment, we have also newly performed experiments for the scenerios where the data is more sparse (sampling period of 0.1, 0.3, 0.5, 0.7), and the noise is stronger (noise amplitude of 5, 10, 20, 50% of the mean). The results show that our algorithm is very robust to increased sparsity, but less so for the cases when the noise amplitude exceeds 20% of the mean. We attribute this to the deterioration of the spline interpolation used. However, we believe that this is not a big limitation as this degradation of the interpolant can be countered by increasing the degree of smoothing used to construct the interpolant.
>
> Overall, we want to thank you for your detailed constructive comments. While we were not able to produce the results in time, your comments have been pivotal in us improving our work and updating our preprint.

---

### Decision · Program_Chairs · 2023-01-20

**Decision:**

Reject

**Justification For Why Not Higher Score:**

see above.

**Justification For Why Not Lower Score:**

N/A

**Metareview: Summary, Strengths And Weaknesses:**

All reviewers have raised major concerns with this paper, and the authors have not responded. A clear reject.